# Genetic and Distribution Data of the Bramble Shark *Echinorhinus brucus* (Bonnaterre, 1788) and the Prickly Shark *Echinorhinus cookei* Pietschmann, 1928 to Better Reconstruct Their Conservation Status

**DOI:** 10.3390/ani14070993

**Published:** 2024-03-23

**Authors:** Matteo Battiata, Fabrizio Serena, Sabrina Lo Brutto

**Affiliations:** 1Department Earth and Marine Sciences (DiSTeM), University of Palermo, Via Archirafi 20, 90123 Palermo, Italy; matteo.battiata@unipa.it; 2NBFC, National Biodiversity Future Center, Piazza Marina 61, 90133 Palermo, Italy; 3National Research Council—Institute of Marine Biological Resources and Biotechnology, Via Vaccara 61, 91026 Mazara del Vallo, Italy; fabrizio50serena@gmail.com

**Keywords:** biodiversity assessment, bramble shark, prickly sharks, *Echinorhinus brucus*, *E. cookei*, genetic variability, conservation status

## Abstract

**Simple Summary:**

Sharks and rays are threatened by human activities like fishing and climate change. Many of these species are at risk of extinction. Despite efforts, we still do not know enough about the biology of some species. This lack of information is often correlated to inaccessible habitats, as in the case of the Bramble shark, *Echinorhinus brucus*, and the Prickly shark, *E. cookei*, which live in deep oceans. *Echinorhinus brucus* and *E. cookei* are the only species of the genus and differ in the shapes and patterns of denticles on their skin. In this study, the data collected demonstrated that *E. brucus* is present worldwide, except in the Pacific Ocean, where *E. cookei* lives. Though there are signs of local extinction of the Bramble shark from the North Sea and the western Mediterranean Sea, two areas with a significantly high number of captures of *E. brucus* are the western Atlantic Ocean and the central Indian Ocean. These areas could be important for conservation plans, especially considering the genetic differences shown between the Bramble shark populations from the Indian Ocean and the Atlantic Ocean. The species could include two different taxonomic entities that have not yet been fully detected by scientists.

**Abstract:**

Elasmobranch species show low resilience in relation to anthropogenic stressors such as fishing efforts, loss of habitats, and climate change. In this sense, the elasmobranch populations appear to be at risk of extinction in many cases. Despite conservation researchers making efforts to implement knowledge, the information on the biology, reproduction, distribution, or genetic structure of some species is still scattered, often caused by the occurrence of species in inaccessible habitats. *Echinorhinus brucus* is a deep benthic shark evaluated as “Endangered” on which little information is available, particularly about its geographical range and genetic structure, while *E. cookei* is listed as “Data Deficient”. *Echinorhinus brucus* belongs to the Echinorhinidae family, and its unique congeneric species is *E. cookei*. The main morphological diagnostic characteristic of both species is the presence of denticles with different shapes and patterns on the derma. In the present paper, mitochondrial COI and NADH2 sequences were retrieved from both *E. brucus* and *E. cookei* species, and analyses were conducted by applying different models of phylogenetic inference. Sequences of *E. brucus* captured in the Indian Ocean (IOS) did not cluster with the Atlantic *E. brucus* counterparts (AOS) but instead with *E. cookei* sequences; the different models showed an overlapping tree topology. Concurrently, a review of the historical and recent captures of the two species was carried out. The worldwide distribution of *E. brucus* excludes the Pacific Ocean area, where *E. cookei* occurs, and is characterised by presumably current local extinctions in the North Sea and the western Mediterranean Sea. The dataset describes two definite areas of significantly high abundance of *E. brucus* located in the Atlantic Ocean (Brazil) and the Indian Ocean (India). These areas suggest zones for conservation plans, especially considering the two lineages identified through molecular approaches.

## 1. Introduction

The IUCN list of endangered and threatened marine species is constantly updated, and new species are being added with increasing fishing pressure and climate change [1]. Despite the efforts of scientists engaged in conservation challenges, knowledge of the status of many species is still lacking [2]. The scarcity of the biology, ethology, reproduction, distribution, or genetics data makes it difficult to assign a correct IUCN risk category and plan interventions for conservation.

Although substantial efforts have been dedicated to the study of elasmobranchs, a notable subset of species within this group remains in need of further investigation: the Bramble shark *Echinorhinus brucus* (Bonnaterre, 1788) (Echinorhiniformes, Echinorhinidae) is one of these. Regrettably, our understanding of this species is severely constrained, and data about it are not often updated and well arranged; only a few reviews and specimen checklists have been performed during the last decade [3]. Individuals of this species have rarely been fished worldwide in the nineteenth century and captured as bycatch or target species in meat and liver oil fisheries [4]. A source of information on the species is natural history museums or the sighting records collected in different ways [5,6,7].

*Echinorhinus brucus* is a deep benthic shark (more frequent in the 200–900 m range) [8,9,10] that prefers soft-bottom habitats of the continental slope, but some records also refer to catches in very shallow waters [11,12]. In *E. brucus*, maturity occurs at 189–231 cm for females and 150–187 for males [11,13,14,15,16]. Its diet consists mainly of crustaceans (69%) and teleosteans (26%), together with cephalopods (1.7%) and elasmobranchs (0.7%) [17]. Its presence is confirmed in a wide geographical area, from the North Atlantic coasts to the South American ones and in the Atlantic coasts of Europe and Africa, extending to the Mediterranean Sea and the Indian Ocean [4,10,11,17,18,19,20], though the global distribution seems to be particularly fragmented [21]. A significant reduction in the number of individuals in fisheries bycatch has been recorded in the Mediterranean Sea, and it is assumed that the abundance of this species may suffer a sharp reduction (80–100%) by the end of the century due to depth fishing [21,22]. In particular, the most recent catches (2002–2013) in the Mediterranean Sea were concentrated in the Levantine region [23,24]. As for reproduction, *E. brucus* is a viviparous species, with 10 and 52 young in a litter [11,25]; the dimensions at birth are between 40 and 55 cm, and no reproductive seasonality has been observed [11].

*Echinorhinus brucus* belongs to the Echinorhinidae family in addition to their congeneric species, *Echinorhinus cookei* Pietschmann, 1928, also known as the prickly shark. In both species, their conservation is concerned with bycatch events in bottom trawls and line gear because they are sometimes caught during fisheries of commercial interest products [17,26]. The main diagnostic characteristic between the two species is the morphological pattern of dermal denticles. In *E. brucus*, the denticles have various sizes, spaced on the body, and some of them fused in groups; on the contrary, in *E. cookei* denticles are relatively small (less than 5 mm), stellate bases, close together, and with denticles not expanded into large bucklers or thorns [10,27]. *Echinorhinus cookei* seems to be present up to greater depths (1100 m) [25] and may reach greater maximum dimensions (400 cm). Its distribution seems to be pan-Pacific and in a bordering area of the eastern Indian Ocean, along Australian coasts [15,27]; in the eastern Pacific Ocean, the distribution seems to be continuous, from Oregon (USA) to Chile [28]. Recently, an Atlantic capture presented questionable identification. In 2012, a female was captured in the Caribbean waters of Venezuela, exhibiting a particular pattern of dermal denticles that did not allow a confident identification at the species level due to the shape intermediate between *E. brucus* and *E. cookei* [29]. The definitive identification occurred only after molecular analysis attributing the individual to *E. brucus* [29].

On the whole, the genetic variability of both species has never been deeply investigated; as such, the phylogenetic relationship of *E. brucus* vs. *E. cookei* is poorly known. Only a few studies have outlined the *Echinorhinus* genus genetic pattern, which seemed to show incongruity with actual valid taxonomy [30,31,32].

In the present paper, we extrapolated phylogenetic tree topology from two molecular markers of existing sequences using much more robust evolutionary models to evaluate the congruence with the taxonomy of the genus and presented the worldwide historical and current distribution of the two species using updated literature.

## 2. Materials and Methods

A first bibliography analysis was conducted to locate and quantify the molecular markers available. The mitochondrial NADH dehydrogenase subunit 2 (NADH2) and cytochrome c oxidase subunit I (COI) sequences from *Echinorinus* specimens were downloaded from Bold System [33] and NCBI [34]. BLAST research [35] was conducted using default parameters to verify the taxonomic accuracy. A total of six sequences attributed to *E. brucus* and seven attributed to *E. cookei* were found in databases for the COI marker, while five and one sequences were found, respectively, for *E. brucus* and *E. cookei* with regard to the NADH2 marker. Though the NADH2 sequences from Sri Lanka were uploaded on Genbank as *Echinorhinus* sp., as reported by the authors, we verified through pictures of individuals that the morphological features of the samples were consistent with *E. brucus* [32]. Similarly, the NADH2 sequence of an individual from Oman was allocated by authors as *E. brucus* species, according to its morphology [30,31,36]. Three sequences of *Pristiophorus cirratus* (Latham, 1974) and one sequence of *Squalus crassispinus* Last, Edmunds & Yearsley, 2007, were used as outgroups, respectively, for COI and NADH2 analysis. Sequences were handled on BioEdit and aligned with ClustalW multiple alignment options using default parameters [37]. A complete genome sequence of *E. cookei* was used in both NADH2 and COI analysis; it was aligned in both databases and cut with a length of a 670 base pair for COI and 1047 bp for NADH2.

Phylogenetic Maximum Likelihood (ML) and Bayesian Inference (BI) analyses were conducted to search for coherence among trees for both molecular markers.

The ML analyses were conducted for both markers on IQ3 online portal [38] using default settings. Two ML analyses for each marker were carried out, one for nucleotide sequences and one for amino acid sequences. The amino acids dataset was created by translating the sequences of NADH2 on MEGA version 11 [39] using the alignment session and the Vertebrate Mitochondrial Genetic Code.

All Bayesian Inference analyses were conducted only for COI marker on CIPRES Science Gateway [40] using Beast2 2.6.6; the input files for all analyses were handled using BEAUTi2 2.6.0 [41,42].

Three different Bayesian Inference analyses were performed to see if differences were presented in the tree topologies between three different models. In all analyses, the substitution rate found by Martin et al. [43] and Dudgedon et al. [44] was used for the calibration of the tree. The HKY model was chosen as the best result from ML autodetected output. Settings were as follows: HKY gamma 4 model, normal relaxed clock log, clock rate, 0.07 × 10^−8^ substitution per year, and 100,000.000 generations, sampled every 10,000 for Birth and Death model, Yule model, and Coalescent model.

The p-distance of COI was calculated in MEGA version 11 [39] using the Gamma Distributed Rates among Site and Gamma parameter 4.00, and results are shown in Table 1.

Regarding the distribution dataset, a deep bibliographic review was conducted analysing all bycatches events, museum specimens [6], results of projects [45], and any sighting already published for both *E. brucus* and *E. cookei* species [16,17,19,23,26,28,29,30,31,32,36,46,47,48,49,50,51,52,53,54,55,56,57,58,59,60,61,62,63,64,65,66,67,68,69,70]; all data are grouped in old captures (before 1930) and recent captures (after 1930), listed in Table 2, and displayed in a world map (Figure 1).

## 3. Results

The COI and NADH2 nucleotides and NADH2 amino acid sequences results from Maximum Likelihood (ML) analyses showed the same tree topology. In the phylogenetic trees of both markers, *E. brucus* sequences show a non-monophyletic topology. The Indian Ocean Sequences (IOS) of *E. brucus* (i.e., samples from Indian and Oman coasts for COI and Sri Lanka and Oman coasts for NADH2) did not cluster with the Atlantic Ocean Sequences (AOS) (i.e., samples from Venezuelan and USA coasts for COI and USA coasts for NADH2) but discriminated into two different lineages. The Indian (IOS) lineage resulted in being a sister group of *E. cookei*. Only the amino acid ML tree for NADH2 showed a significative bootstrap value at the node of splitting IOS *E. brucus* and *E. cookei* (shown in Figure 2). For this node, the bootstrap value was 60 in the COI nucleotide ML tree (tree not shown). No evolutionary signal was present in the COI amino acid result.

All COI trees built with BI (Bayesian Inference) analysis showed the same topology of ML trees, but in this case, all of them showed weak nodes between IOS *E. brucus* and *E. cookei*, ranging in a posterior value of 50–57 (the Bayesian tree using the Yule model is shown in Figure 3).

It is noteworthy to highlight that the nodes at the root of each of the three major lineages (AOS *E. brucus*–IOS *E. brucus–E. cookei*) were highly significative (99–100) in all the trees, a sign that these lineages are well differentiated from each other (Figure 2 and Figure 3).

The COI pattern for *E. cookei* (Figure 3), which included a higher number of sequences than the NADH analysis, showed that haplotypes were divided into two subgroups. This was not consistent with a geographical outline. In fact, the sequences from Australia did not cluster together. The two specimens from Tasmania and Queensland were separated from two sequences from New South Wales and Tasmania.

The p-distance ranges between 0.0368 and 0.0449 when comparing individuals from the three major groups and is around 10 times greater than the internal difference of the groups (0.000–0.0077) (Table 1).

The global geographical distribution of *E. brucus* and *E. cookei*, extrapolated from captures and sightings reported in the literature, is presented in Table 2 and Figure 1. The map of distribution discriminates specimens that were found before 1930 and from 1930 to the present, as well as the range of the number of individuals observed in a specific area. The total number of specimens herein listed is 8903 for *E. brucus* and 301 for *E. cookei*, respectively, with an average number of 37 (*E. brucus*) and 5 (*E. cookei*) specimens per locality in a period comprising 1680 to 2022 (*E. brucus*) and 1956 to 2022 (*E. cookei*).

## 4. Discussion

In the present paper, different models of phylogenetic inference, applied to mitochondrial sequences, described a pattern where *E. brucus* from the Indian Ocean (IOS) did not cluster with the Atlantic *E. brucus* lineage (AOS), but instead clustered with *E. cookei* sequences.

Although the results could be affected by the small number of data, our analyses support the taxonomic criticisms within the genus that have already been noticed by other authors [29,30,31,32,69]. In particular, Fernando et al. [32] and Naylor et al. [31] exhibited the same topology of the present trees using different methodologies. The first authors used the Neighbor-Joining method with the uncorrected p-distance model, and the second used a Bayesian approach using the GTR+I+Γ model, both using NADH2 as a marker.

The COI is usually a good marker for species and subspecies delimitation because of its peculiarity to be a fast mutational rate gene [74,75]; nevertheless, the mutational rate of NADH2 in *Echinorhinus* sharks seems to be even faster than COI in the first position of the codon. This leads to a higher incidence of mutations in the amino acid sequences compared to COI, where a greater prevalence of silent mutations is generally observed [30,31,76,77].

The great phylogenetic distance (Table 1) shown in the trees between the two *E. brucus* lineages, confirmed by both molecular markers, may be explained as a probable separation due to distance or barriers to gene flow between the two main clusters (IOS vs. AOS). In fact, *E. brucus* is widespread; in the Atlantic Ocean, some records occur at high latitudes (Virginian coasts and the North Sea) and along the coast of Chubut in Argentina and South Africa [6,12,46,50], and presumably in almost all of the Indian Ocean (Figure 1, Table 2). Similar patterns are observable for other elasmobranchs widely distributed, where distal populations have been demonstrated to be genetically heterogeneous [78].

The dataset of Table 2 and the distribution map (Figure 1) show two main areas of aggregations, detected by the highest values of captures (thousands of individuals) in Brazilian and Indian waters. The two areas are congruent with the separation of the two lineages and should be carefully investigated. There is no evidence of the presence of *E. brucus* in the Pacific Ocean, corroborated by the sole presence of *E. cookei* (Table 2). The highest number of records for *E. cookei* occurs from northern California to Costa Rica, Chile, and Cocos Island, where, despite the inaccessibility of the species’ habitats, some direct sightings occurred during submersible missions [54,63]. The latter are the only sightings of living animals in their habitat for the *Echinorhinus* genus. The capture of an *E. cookei* individual in Indonesia (Figure 1; Table 2) is the westernmost point for this species; this is the unique specimen from the Indian Ocean and the only record of an overlapping area between the two species’ ranges [65].

Molecular results for *E. cookei* (Figure 3) show that individuals from Australia do not cluster together. The two Australian specimens, from Tasmania and Queensland, are strictly grouped with the one from Hawaii. The other two, from New South Wales and Tasmania, cluster with individuals from Chile and Costa Rica. The actual knowledge of the reproductive strategies of *E. cookei* is insufficient to assert any consideration, but this could serve as a stimulus for new studies; the possible dispersal behaviour of this species, old biogeographical relations, or presence of independent mutations may have affected the population or the results.

Regarding *E. brucus*, hotspots of bycatch occurrences can be assumed to be in the presence of upwelling events where aggregation phenomena have occurred in the same localities over the years [51]. A similar pattern is also observed in *E. cookei* [26,54,63] (Table 2). On the contrary, *E. brucus* is now rarely fished in some regions. Although the dataset of the distribution (Table 2, Figure 1) shows regions where the captures reach very high abundance for both species, a new report highlights how Indian *E. brucus* populations suffer from excessive fishing pressure due to the high demand for liver oil, which caused a reduction in landings in the last 10 years [4]. Extensive studies are needed to program and manage any kind of conservation action because, as we have shown, reproductive isolation occurs within the species, and it has not yet been fully researched.

## 5. Conclusions

*Echinorhinus brucus* and *E. cookei* remain poorly known and enigmatic species.

A great divergence between *E. brucus* Indian (IOS) lineage and *E. brucus* Atlantic (AOS) lineage was confirmed by our analyses, conducted with different methodologies (Maximum Likelihood and Bayesian Inference). The two lineages were further shown to be paraphyletic. Thus, a precautional revision of the species and the genus needs, at least, subspecies identification and correct information in support of conservation plans. In order to better reconstruct the unclear taxonomy of *E. brucus* and *E. cookei*, a morphological and genetic revision on ancient and modern specimens should be carried out.

Individuals of *Echinorhinus* genus are now rarely fished in some regions of the world, except in some areas where they contribute significantly to the overall biomass as bycatch product to be handled. *E. brucus* disappeared from the western Mediterranean Sea and North Sea, while it is very abundant in the Indian Ocean and Southwestern Atlantic Ocean. *E. cookei*’s distributional ancient data do not exist, and a comparative analysis between present and past abundance is partial or unworkable. In any case, for both species, some regions present a significant number of records. Extensive studies are thus needed to program and manage any kind of conservation action, placing particular attention on the abundance zones for their relevance. From this point of view, the deep genetic divergence within *E. brucus* and the evolutive relation with the congeneric *E. cookei*, yet to be clarified, should be used to set a baseline for future actions.

## Figures and Tables

**Figure 1 animals-14-00993-f001:**
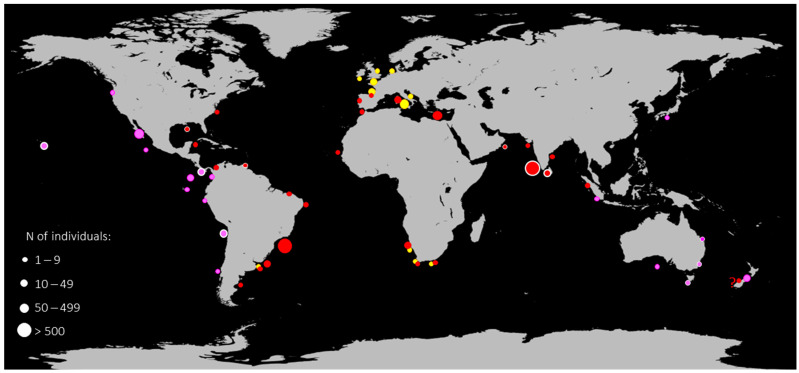
Map of distribution of *E. brucus* and *E. cookei* extrapolated from captures and sightings; data are listed in detail in Table 2. Yellow dots indicate *E. brucus* specimens before 1930; red dots *E. brucus* specimens from 1930 to present; purple dots specimens of *E. cookei*. The dot dimension indicates the range of the number of individuals observed in a specific area. Dots circled with white lines indicate the areas from where molecular data have been used in the present study. The dot with “?” indicates dubious sightings of two *E. brucus* individuals in New Zealand waters [61].

**Figure 2 animals-14-00993-f002:**
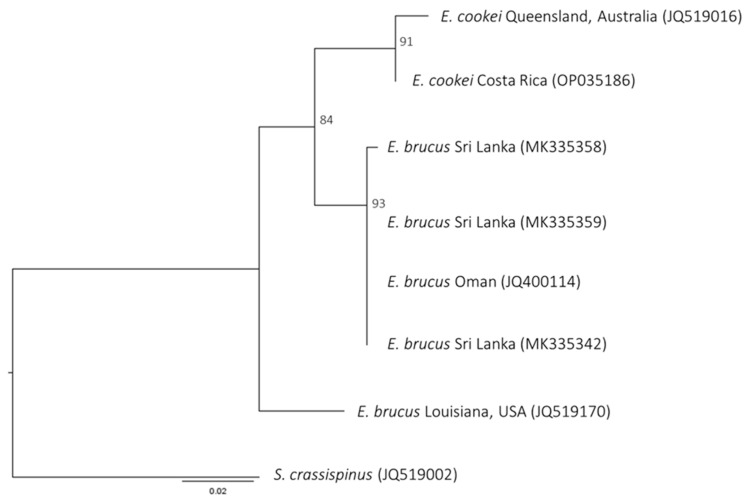
Maximum Likelihood tree based on NADH2 amino acid sequences; all nodes show significant bootstrap values.

**Figure 3 animals-14-00993-f003:**
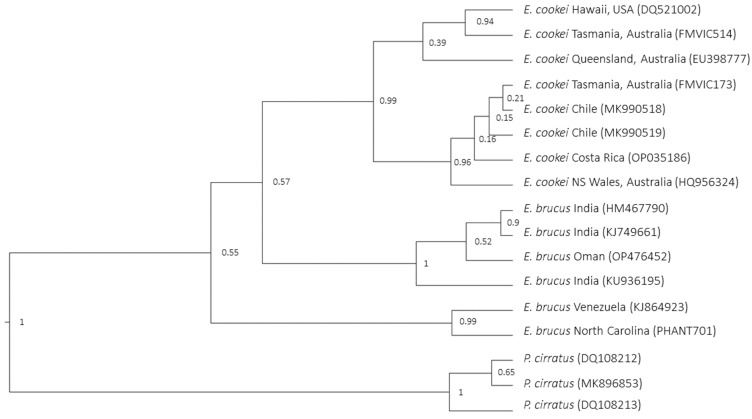
Bayesian analysis tree using Yule model for COI marker. The node that delineates the cluster of IOS *E. brucus* + *E. cookei* exhibits a low posterior value, as does the innermost node clustering the AOS *E. brucus*. On the contrary, the nodes at the root of the three separate lineages (respectively, AOS *E. brucus*, IOS *E. brucus*, and *E. cookei*) show a very high posterior value.

**Table 1 animals-14-00993-t001:** COI p-distance of *E. brucus* and *E. cookei* sequences; the great difference in the three major groups is notable (IOS *E. brucus*, AOS *E. brucus*, and *E. cookei*).

	Species	1	2	3	4	5	6	7	8	9	10	11	12	13	14
1	*E. brucus* India (KJ749661)														
2	*E. brucus* India (KU936195)	0.0066													
3	*E. brucus* India (HM467790)	0.0000	0.0066												
4	*E. brucus* Oman (OP476452)	0.0033	0.0032	0.0032											
5	*E. brucus* USA (PHANT701-08)	0.0373	0.0378	0.0373	0.0383										
6	*E. brucus* Venezuela (KJ864923)	0.0374	0.0377	0.0372	0.0382	0.0048									
7	*E. cookei* Chile (MK990518)	0.0374	0.0410	0.0372	0.0398	0.0417	0.0368								
8	*E. cookei* Hawaii (DQ521002)	0.0390	0.0426	0.0388	0.0414	0.0449	0.0399	0.0061							
9	*E. cookei* Australia NSW (HQ956324)	0.0391	0.0429	0.0391	0.0417	0.0417	0.0386	0.0015	0.0077						
10	*E. cookei* Australia Queensland (EU398777)	0.0407	0.0443	0.0405	0.0430	0.0433	0.0383	0.0061	0.0092	0.0077					
11	*E. cookei* Australia Tasmania (FMVIC173-08)	0.0374	0.0410	0.0372	0.0398	0.0417	0.0368	0.0000	0.0061	0.0015	0.0061				
12	*E. cookei* Australia Tasmania (FMVIC514-08)	0.0374	0.0410	0.0372	0.0398	0.0433	0.0383	0.0046	0.0046	0.0062	0.0077	0.0046			
13	*E. cookei* Costa Rica (OP035186)	0.0390	0.0417	0.0388	0.0405	0.0433	0.0382	0.0015	0.0077	0.0031	0.0077	0.0015	0.0061		
14	*E. cookei* Chile (MK990519)	0.0374	0.0410	0.0372	0.0398	0.0417	0.0368	0.0000	0.0061	0.0015	0.0061	0.0000	0.0046	0.0015	

**Table 2 animals-14-00993-t002:** *Echinorhinus brucus* and *E. cookei* specimens captured or sighted, ordered by years from past to present. Marine area, locality, number of individuals, and reference, NCBI Accession Number (A.N.) or Bold System code, are included. The total dataset is shown in Figure 1, except for samples for which geographical information was not available (NA).

Year	Ocean or Sea	Locality	Species	N.ind	References
1680	Atlantic Ocean	Bayonne, France	*E. brucus*	1	Mollen & Iglésias, 2023 [6]
1798	Mediterranean Sea	Nice, France	*E. brucus*	1	Mancusi et al., 2020 [45]
1827	Mediterranean Sea	Genoa, Italy	*E. brucus*	1	Mollen & Iglésias, 2023 [6]
1834	NA	NA	*E. brucus*	1	Mollen & Iglésias, 2023 [6]
1837	Atlantic Ocean	Cape of Good Hope, South Africa	*E. brucus*	1	Mollen & Iglésias, 2023 [6]
1837	Atlantic Ocean	Brixham, Devon, UK	*E. brucus*	1	Mollen & Iglésias, 2023 [6]
1838	Atlantic Ocean	Cape of Good Hope, South Africa	*E. brucus*	1	Mollen & Iglésias, 2023 [6]
1848	Atlantic Ocean	Lisbon, Portugal	*E. brucus*	1	Mollen & Iglésias, 2023 [6]
1853	Atlantic Ocean	Cape of Good Hope, South Africa	*E. brucus*	1	Mollen & Iglésias, 2023 [6]
1856	Mediterranean Sea	Nice, France	*E. brucus*	1	Mollen & Iglésias, 2023 [6]
1856	Mediterranean Sea	Nice, France	*E. brucus*	1	Mancusi et al., 2020 [45]
1860	Mediterranean Sea	Nice, France	*E. brucus*	1	Mollen & Iglésias, 2023 [6]
1861	Mediterranean Sea	Nice, France	*E. brucus*	1	Mollen & Iglésias, 2023 [6]
1864	Atlantic Ocean	Charente-Maritime, France	*E. brucus*	1	Mollen & Iglésias, 2023 [6]
1865	Atlantic Ocean	Cornwall, UK, English Channel	*E. brucus*	1	Mollen & Iglésias, 2023 [6]
1865	Mediterranean Sea	NA	*E. brucus*	1	Mollen & Iglésias, 2023 [6]
1865	Mediterranean Sea	Nice, France	*E. brucus*	3	Mollen & Iglésias, 2023 [6]
1866	Atlantic Ocean	Portugal	*E. brucus*	1	Mollen & Iglésias, 2023 [6]
1867	Atlantic Ocean	Polperro, Cornwall, UK, English Channel	*E. brucus*	1	Mollen & Iglésias, 2023 [6]
1867	Mediterranean Sea	Nice, France	*E. brucus*	1	Mollen & Iglésias, 2023 [6]
1868	Mediterranean Sea	Tyrrhenian Sea, Italy	*E. brucus*	3	Mollen & Iglésias, 2023 [6]
1868	Atlantic Ocean	Bo’ness, Linlithgowshire, Scotland	*E. brucus*	1	Mollen & Iglésias, 2023 [6]
1869	Mediterranean Sea	France	*E. brucus*	1	Mollen & Iglésias, 2023 [6]
1870	Mediterranean Sea	Nice, France	*E. brucus*	1	Mollen & Iglésias, 2023 [6]
1870	Mediterranean Sea	Palermo, Tyrrhenian Sea, Italy	*E. brucus*	1	Mollen & Iglésias, 2023 [6]
1870	Mediterranean Sea	Tyrrhenian Sea, Italy	*E. brucus*	4	Mollen & Iglésias, 2023 [6]
1870	Mediterranean Sea	Palermo, Tyrrhenian Sea, Italy	*E. brucus*	1	Mancusi et al., 2020 [45]
1871	Mediterranean Sea	Tyrrhenian Sea, Italy	*E. brucus*	1	Mollen & Iglésias, 2023 [6]
1871	Atlantic Ocean	Earlsferry, Elie, Scotland, Northern Sea	*E. brucus*	1	Mollen & Iglésias, 2023 [6]
1872	Atlantic Ocean	Ireland	*E. brucus*	1	Mollen & Iglésias, 2023 [6]
1872	Atlantic Ocean	Brixham, Devon, UK, English Channel	*E. brucus*	1	Mollen & Iglésias, 2023 [6]
1872	Atlantic Ocean	Cornwall, UK, English Channel	*E. brucus*	1	Mollen & Iglésias, 2023 [6]
1872	Mediterranean Sea	Gulf of Fréjus, France	*E. brucus*	1	Mollen & Iglésias, 2023 [6]
1872	Mediterranean Sea	Palermo, Tyrrhenian Sea, Italy	*E. brucus*	1	Mancusi et al., 2020 [45]
1872	Mediterranean Sea	Palermo, Tyrrhenian Sea, Italy	*E. brucus*	1	Mollen & Iglésias, 2023 [6]
1874	Mediterranean Sea	Palermo, Tyrrhenian Sea, Italy	*E. brucus*	1	Mancusi et al., 2020 [45]
1874	Mediterranean Sea	Palermo, Tyrrhenian Sea, Italy	*E. brucus*	1	Mollen & Iglésias, 2023 [6]
1874	Atlantic Ocean	Yorkshire, UK, Northern Sea	*E. brucus*	1	Mollen & Iglésias, 2023 [6]
1875	Atlantic Ocean	Cornwall, UK, English Channel	*E. brucus*	1	Mollen & Iglésias, 2023 [6]
1875	Atlantic Ocean	Cornwall, UK, English Channel	*E. brucus*	1	Mollen & Iglésias, 2023 [6]
1875	Atlantic Ocean	Aberdeen, Scotland, Ythan River estuary	*E. brucus*	1	Mollen & Iglésias, 2023 [6]
1876	Mediterranean Sea	Kvarner Gulf, Croatia	*E. brucus*	1	Mancusi et al., 2020 [45]
1876	Mediterranean Sea	Livorno, Italy	*E. brucus*	1	Mancusi et al., 2020 [45]
1876	Mediterranean Sea	Livorno, Italy	*E. brucus*	1	Mollen & Iglésias, 2023 [6]
1876	Mediterranean Sea	Nice, France	*E. brucus*	1	Mollen & Iglésias, 2023 [6]
1877	Mediterranean Sea	Gulf of Kvarner, Croatia, Adriatic Sea	*E. brucus*	1	Mollen & Iglésias, 2023 [6]
1879	Mediterranean Sea	Nice, France	*E. brucus*	1	Mancusi et al., 2020 [45]
1879	Mediterranean Sea	Nice, France	*E. brucus*	1	Mollen & Iglésias, 2023 [6]
1872–1880	Mediterranean Sea	Tyrrhenian Sea, Italy	*E. brucus*	1	Mollen & Iglésias, 2023 [6]
1880	Atlantic Ocean	Concarneau, France	*E. brucus*	1	Mollen & Iglésias, 2023 [6]
1880	Mediterranean Sea	Ligurian Sea, Italy	*E. brucus*	1	Mollen & Iglésias, 2023 [6]
1880	Mediterranean Sea	Marseille, France	*E. brucus*	1	Mollen & Iglésias, 2023 [6]
1882	Atlantic Ocean	Le Croisic, Loire, France	*E. brucus*	1	Mollen & Iglésias, 2023 [6]
1884	Atlantic Ocean	Biarritz, France	*E. brucus*	1	Mollen & Iglésias, 2023 [6]
1884	Atlantic Ocean	Walvis Bay, Namibia	*E. brucus*	2	Mollen & Iglésias, 2023 [6]
1884	Mediterranean Sea	Nice, France	*E. brucus*	1	Mollen & Iglésias, 2023 [6]
1884	NA	NA	*E. brucus*	1	Mollen & Iglésias, 2023 [6]
1884	NA	NA	*E. brucus*	1	Mollen & Iglésias, 2023 [6]
1885	Atlantic Ocean	Hag’s Head, W Ireland	*E. brucus*	1	Mollen & Iglésias, 2023 [6]
1885	Mediterranean Sea	Tuscany, Italy	*E. brucus*	1	Mollen & Iglésias, 2023 [6]
1887	Mediterranean Sea	Genova, Italy	*E. brucus*	1	Mancusi et al., 2020 [45]
1887	Mediterranean Sea	NA	*E. brucus*	1	Mollen & Iglésias, 2023 [6]
1887	Mediterranean Sea	Genova, Italy	*E. brucus*	1	Mollen & Iglésias, 2023 [6]
1889	Atlantic Ocean	Buarcos, Coimbra, Portugal	*E. brucus*	1	Mollen & Iglésias, 2023 [6]
1891	Mediterranean Sea	Nice, France	*E. brucus*	1	Mollen & Iglésias, 2023 [6]
1891	Mediterranean Sea	Nice, France	*E. brucus*	1	Mollen & Iglésias, 2023 [6]
1892	Atlantic Ocean	Europe	*E. brucus*	1	Mollen & Iglésias, 2023 [6]
1893	Atlantic Ocean	Galloper Bank, Suffolk, UK, Northern Sea	*E. brucus*	2	Mollen & Iglésias, 2023 [6]
1894	NA	NA	*E. brucus*	1	Mollen & Iglésias, 2023 [6]
1895	Mediterranean Sea	Nice, France	*E. brucus*	1	Mollen & Iglésias, 2023 [6]
1896	Atlantic Ocean	Plymouth, Devon, UK	*E. brucus*	1	Mollen & Iglésias, 2023 [6]
1896	Atlantic Ocean	SW Ireland	*E. brucus*	1	Mollen & Iglésias, 2023 [6]
1897	Atlantic Ocean	Setubal, Portugal	*E. brucus*	1	Mollen & Iglésias, 2023 [6]
1898	Atlantic Ocean	Bay of Biscay, France	*E. brucus*	1	Mollen & Iglésias, 2023 [6]
1898	Atlantic Ocean	Portugal	*E. brucus*	1	Mollen & Iglésias, 2023 [6]
1898	Atlantic Ocean	Buenos Aires, Argentina	*E. brucus*	1	Mollen & Iglésias, 2023 [6]
1898	Atlantic Ocean	Cornwall, UK, English Channel	*E. brucus*	1	Mollen & Iglésias, 2023 [6]
1898	Mediterranean Sea	Nice, France	*E. brucus*	1	Mancusi et al., 2020 [45]
1898	Mediterranean Sea	Nice, France	*E. brucus*	1	Mollen & Iglésias, 2023 [6]
1898	NA	UK	*E. brucus*	1	Mollen & Iglésias, 2023 [6]
1900	Atlantic Ocean	Mossel Bay, South Africa	*E. brucus*	1	Mollen & Iglésias, 2023 [6]
1902	Atlantic Ocean	UK	*E. brucus*	1	Mollen & Iglésias, 2023 [6]
1902	Atlantic Ocean	Bournemouth, Dorset, UK	*E. brucus*	1	Mollen & Iglésias, 2023 [6]
1902	Mediterranean Sea	Nice, France	*E. brucus*	2	Mollen & Iglésias, 2023 [6]
1903	Atlantic Ocean	Hamburg, Northern Sea	*E. brucus*	2	Mollen & Iglésias, 2023 [6]
1904	Mediterranean Sea	Chioggia, Adriatic Sea, Italy	*E. brucus*	1	Mollen & Iglésias, 2023 [6]
1904	Mediterranean Sea	Chioggia, Italy	*E. brucus*	1	Mancusi et al., 2020 [45]
1908	Mediterranean Sea	Italy	*E. brucus*	1	Mollen & Iglésias, 2023 [6]
1908	NA	UK	*E. brucus*	1	Mollen & Iglésias, 2023 [6]
1909	Atlantic Ocean	Eastern Cape, South Africa	*E. brucus*	1	Mollen & Iglésias, 2023 [6]
1909	Mediterranean Sea	Napoli, Italy	*E. brucus*	1	Mancusi et al., 2020 [45]
1909	Atlantic Ocean	E Isle of May, Scotland, Northern Sea	*E. brucus*	1	Mollen & Iglésias, 2023 [6]
1910	Mediterranean Sea	Nice, France	*E. brucus*	1	Mollen & Iglésias, 2023 [6]
1912	Atlantic Ocean	Bay of Biscay, France	*E. brucus*	1	Mollen & Iglésias, 2023 [6]
1919	Atlantic Ocean	San Sebastian, BAC, Spain	*E. brucus*	1	Mollen & Iglésias, 2023 [6]
1922	Atlantic Ocean	South Africa	*E. brucus*	1	Mollen & Iglésias, 2023 [6]
1922	Atlantic Ocean	Western Cape, South Africa	*E. brucus*	5	Mollen & Iglésias, 2023 [6]
1923	Mediterranean Sea	Ligurian Sea, Italy	*E. brucus*	1	Mancusi et al., 2020 [45]
1923	Mediterranean Sea	Ligurian Sea, Italy	*E. brucus*	1	Mollen & Iglésias, 2023 [6]
1923	Mediterranean Sea	Noli, Italy	*E. brucus*	1	Mancusi et al., 2020 [45]
1923	Mediterranean Sea	Noli, Ligurian Sea, Italy	*E. brucus*	1	Mollen & Iglésias, 2023 [6]
1925	Atlantic Ocean	Cantabria, Spain	*E. brucus*	3	Mollen & Iglésias, 2023 [6]
1925	Atlantic Ocean	Charente-Maritime, France	*E. brucus*	1	Mollen & Iglésias, 2023 [6]
1933	Mediterranean Sea	Nice, France	*E. brucus*	1	Mollen & Iglésias, 2023 [6]
1934	Mediterranean Sea	Palermo, Italy	*E. brucus*	1	Mancusi et al., 2020 [45]
1934	Mediterranean Sea	Palermo, Tyrrhenian Sea, Italy	*E. brucus*	1	Mollen & Iglésias, 2023 [6]
1937	Mediterranean Sea	Messina Strait	*E. brucus*	1	Mancusi et al., 2020 [45]
1940	Mediterranean Sea	Golfe d’Aigues-Mortes	*E. brucus*	1	Mancusi et al., 2020 [45]
1940	NA	NA	*E. brucus*	1	Mollen & Iglésias, 2023 [6]
1947	Atlantic Ocean	Pays Basque, France	*E. brucus*	1	Mollen & Iglésias, 2023 [6]
1951	Mediterranean Sea	Camogli, Italy	*E. brucus*	1	Mancusi et al., 2020 [45]
1953	Mediterranean Sea	Camogli, Italy	*E. brucus*	1	Mancusi et al., 2020 [45]
1955	Atlantic Ocean	Mauritania	*E. brucus*	1	Mollen & Iglésias, 2023 [6]
1955	Atlantic Ocean	Rabat, Morocco	*E. brucus*	1	Mollen & Iglésias, 2023 [6]
1955	Atlantic Ocean	Argentina	*E. brucus*	1	Mollen & Iglésias, 2023 [6]
1956	Mediterranean Sea	NA	*E. brucus*	1	Mollen & Iglésias, 2023 [6]
1956	NA	NA	*E. brucus*	1	Mollen & Iglésias, 2023 [6]
1958	Atlantic Ocean	Kayar, Senegal	*E. brucus*	1	Mollen & Iglésias, 2023 [6]
1958	Atlantic Ocean	Region of Thiès, Senegal	*E. brucus*	1	Mollen & Iglésias, 2023 [6]
1958	Atlantic Ocean	Region of Thiès, Senegal	*E. brucus*	1	Mollen & Iglésias, 2023 [6]
1961	Atlantic Ocean	Region of Thiès, Senegal	*E. brucus*	1	Mollen & Iglésias, 2023 [6]
1964	Atlantic Ocean	Eastern Cape, South Africa	*E. brucus*	1	Mollen & Iglésias, 2023 [6]
1965	Atlantic Ocean	Namibia	*E. brucus*	1	Mollen & Iglésias, 2023 [6]
1965	Atlantic Ocean	Western Cape, South Africa	*E. brucus*	1	Mollen & Iglésias, 2023 [6]
1966	Atlantic Ocean	Namibia	*E. brucus*	1	Mollen & Iglésias, 2023 [6]
1968	Atlantic Ocean	La Cotiniere, ile d’Olero, France	*E. brucus*	1	Mollen & Iglésias, 2023 [6]
1968	Atlantic Ocean	Pays Basque, France	*E. brucus*	1	Mollen & Iglésias, 2023 [6]
1968	Atlantic Ocean	Cape Henry, Virginia, USA	*E. brucus*	1	Mollen & Iglésias, 2023 [6]
1968	Atlantic Ocean	Virginia, USA	*E. brucus*	1	Musick & McEachran, 1969
1969	Atlantic Ocean	Namibia	*E. brucus*	1	Mollen & Iglésias, 2023 [6]
1975	Atlantic Ocean	USA	*E. brucus*	1	Mollen & Iglésias, 2023 [6]
1978	Atlantic Ocean	Namibia	*E. brucus*	1	Mollen & Iglésias, 2023 [6]
1978	Atlantic Ocean	Rio Grande do Sul, Brazil	*E. brucus*	1	Mollen & Iglésias, 2023 [6]
1978	Atlantic Ocean	Rio Grande do Sul, Brazil	*E. brucus*	1	Mollen & Iglésias, 2023 [6]
1978	Atlantic Ocean	Rio Grande, Brazil	*E. brucus*	1	Mollen & Iglésias, 2023 [6]
1979	Atlantic Ocean	Spanish–Portuguese coast	*E. brucus*	1	Mollen & Iglésias, 2023 [6]
1960–1980	Atlantic Ocean	Mauritania	*E. brucus*	1	Mollen & Iglésias, 2023 [6]
1960–1980	Atlantic Ocean	NA	*E. brucus*	1	Mollen & Iglésias, 2023 [6]
1960–1980	Atlantic Ocean	NA	*E. brucus*	1	Mollen & Iglésias, 2023 [6]
1980	Atlantic Ocean	Rio Grande do Sul, Brazil	*E. brucus*	1	Mollen & Iglésias, 2023 [6]
1983	Atlantic Ocean	Region of Thiès, Senegal	*E. brucus*	1	Mollen & Iglésias, 2023 [6]
1983	Atlantic Ocean	Namibia	*E. brucus*	1	Mollen & Iglésias, 2023 [6]
1984	Indian Ocean	Kochi Fisheries Harbor, Kerala	*E. brucus*	2	Akhilesh et al., 2013 [17]
1983–1985	Indian Ocean	Kochi Fisheries Harbor, Kerala	*E. brucus*	1	Akhilesh et al., 2013 [17]
1985	Atlantic Ocean	NA	*E. brucus*	1	Mollen & Iglésias, 2023 [6]
1985	Atlantic Ocean	Western Cape, South Africa	*E. brucus*	1	Mollen & Iglésias, 2023 [6]
1985	Atlantic Ocean	Cassino Beach, Rio Grande do Sul, Brazil	*E. brucus*	1	Mollen & Iglésias, 2023 [6]
1985	Mediterranean Sea	Elba (Capo Bianco), Italy	*E. brucus*	1	Mancusi et al., 2020 [45]
1986	Atlantic Ocean	Eastern Cape, South Africa	*E. brucus*	1	Mollen & Iglésias, 2023 [6]
1986	Atlantic Ocean	Tobago	*E. brucus*	1	Mollen & Iglésias, 2023 [6]
1987	Atlantic Ocean	South Africa	*E. brucus*	2	Faure-Beaulieu et al., 2023
1988	Atlantic Ocean	Namibia	*E. brucus*	1	Mollen & Iglésias, 2023 [6]
1988	Atlantic Ocean	Western Cape, South Africa	*E. brucus*	1	Mollen & Iglésias, 2023 [6]
1989	Atlantic Ocean	Louisiana, USA	*E. brucus*	1	Mollen & Iglésias, 2023 [6]
1989	Atlantic Ocean	Louisiana, USA	*E. brucus*	1	Mollen & Iglésias, 2023 [6]
1989	Atlantic Ocean	Rio Grande do Sul, Brazil	*E. brucus*	1	Mollen & Iglésias, 2023 [6]
1989	Atlantic Ocean	Santa Luzia, Brazil	*E. brucus*	1	Mollen & Iglésias, 2023 [6]
1989	Indian Ocean	Kochi Fisheries Harbor, Kerala, India	*E. brucus*	2	Akhilesh et al., 2013 [17]
1991	Atlantic Ocean	Grotto Point, West Coast South Africa	*E. brucus*	1	Mollen & Iglésias, 2023 [6]
1991	Indian Ocean	Kochi Fisheries Harbor, Kerala, India	*E. brucus*	18	Balasubramanian et al., 1993 [48]
1991	Indian Ocean	Tuticorin, India	*E. brucus*	1	Balasubramanian et al., 1993 [48]
1992	Atlantic Ocean	North Carolina, USA	*E. brucus*	1	Mollen & Iglésias, 2023 [6]
1992	Atlantic Ocean	Namibia	*E. brucus*	1	Mollen & Iglésias, 2023 [6]
1992	Atlantic Ocean	North Carolina, USA	*E. brucus*	1	Schwartz, 1993 [49]
1993	Atlantic Ocean	Rio Grande do Sul, Brazil	*E. brucus*	1	Mollen & Iglésias, 2023 [6]
1985–1994	Atlantic Ocean	Namibia	*E. brucus*	1	Mollen & Iglésias, 2023 [6]
1994	Atlantic Ocean	27 km south of Grand Isle Louisiana, USA	*E. brucus*	1	Mollen & Iglésias, 2023 [6]
1994	Atlantic Ocean	NA	*E. brucus*	1	Mollen & Iglésias, 2023 [6]
1994	Atlantic Ocean	Namibia	*E. brucus*	1	Mollen & Iglésias, 2023 [6]
1996	Atlantic Ocean	Namibia	*E. brucus*	1	Mollen & Iglésias, 2023 [6]
1997	Atlantic Ocean	Namibia	*E. brucus*	1	Mollen & Iglésias, 2023 [6]
1997	Atlantic Ocean	Cearà-Bahia, NE Brazil	*E. brucus*	1	Mollen & Iglésias, 2023 [6]
1997	Atlantic Ocean	Chubut, Argentina	*E. brucus*	1	Caille & Olsen, 2000 [50]
1997	Atlantic Ocean	Northeast Brazil	*E. brucus*	1	Ehemann & Zambrano-Vizquel, 2023 [51]
1997	Atlantic Ocean	Northeast Brazil	*E. brucus*	1	Ehemann & Zambrano-Vizquel, 2023 [51]
1997	Atlantic Ocean	Rio Grande do Sul, Brazil	*E. brucus*	1	Mollen & Iglésias, 2023 [6]
1998	Atlantic Ocean	Algarve, Portugal	*E. brucus*	1	Mollen & Iglésias, 2023 [6]
1998	Atlantic Ocean	Cearà-Bahia, NE Brazil	*E. brucus*	1	Mollen & Iglésias, 2023 [6]
1998	Atlantic Ocean	Northeast Brazil	*E. brucus*	1	Ehemann & Zambrano-Vizquel, 2023 [51]
1998	Atlantic Ocean	Northeast Brazil	*E. brucus*	1	Ehemann & Zambrano-Vizquel, 2023 [51]
1998	Atlantic Ocean	Paraiba, Brazil	*E. brucus*	1	Santander-Neto et al., 2022 [52]
2000	Mediterranean Sea	Annaba, Algeria	*E. brucus*	1	Mancusi et al., 2020 [45]
2001	Atlantic Ocean	Rio de Janeiro-Rio Grande, Brazil	*E. brucus*	4378	Perez & Wahrlich, 2005 [26]
2000–2002	Indian Ocean	Kochi Fisheries Harbor, Kerala	*E. brucus*	18	Akhilesh et al., 2013 [17]
2002	Atlantic Ocean	Rio Grande do Sul, Brazil	*E. brucus*	1	Mollen & Iglésias, 2023 [6]
2002	Atlantic Ocean	Rio Grande do Sul, Brazil	*E. brucus*	1	Mollen & Iglésias, 2023 [6]
2002	Indian Ocean	Kochi Fisheries Harbor, Kerala	*E. brucus*	1	Akhilesh et al., 2013 [17]
2002	Mediterranean Sea	Sea of Marmara, Türkiye	*E. brucus*	1	Kabasakal & Bilecenoglu, 2014 [23]
2002	Mediterranean Sea	Türkiye	*E. brucus*	1	Mancusi et al., 2020 [45]
2002	Mediterranean Sea	Easter Algeria	*E. brucus*	1	Hemida & Capapé, 2002 [53]
2004	Atlantic Ocean	Brazil	*E. brucus*	1	Mollen & Iglésias, 2023 [6]
2004	Atlantic Ocean	Rio Grande do Sul, Brazil	*E. brucus*	1	Mollen & Iglésias, 2023 [6]
2004	Indian Ocean	Karachi Fish Harbor, Pakistan	*E. brucus*	1	Moazzam & Osmany, 2021 [71]
2005	Atlantic Ocean	Quintana, Caribbean Sea, Mexico	*E. brucus*	1	Mollen & Iglésias, 2023 [6]
2005	Mediterranean Sea	Aegean Sea, Türkiye	*E. brucus*	1	Kabasakal & Bilecenoglu, 2014 [23]
2005	Mediterranean Sea	Sea of Marmara, Türkiye	*E. brucus*	1	Kabasakal & Bilecenoglu, 2014 [23]
2005	Mediterranean Sea	Türkiye	*E. brucus*	1	Mancusi et al., 2020 [45]
2005	Mediterranean Sea	Türkiye	*E. brucus*	1	Mancusi et al., 2020 [45]
1999/2006	Indian Ocean	Kochi Fisheries Harbor, Kerala, India	*E. brucus*	NA	Sreedhar et al. (2007) [72]
2008	Mediterranean Sea	Sea of Marmara	*E. brucus*	1	Kabasakal & Bilecenoglu, 2014 [23]
2008	Mediterranean Sea	Türkiye	*E. brucus*	1	Mancusi et al., 2020 [45]
2008–2009	Indian Ocean	Kochi Fisheries Harbor, Kerala, India	*E. brucus*	1	Akhilesh et al., 2013 [17]
2009	Indian Ocean	Kochi Fisheries Harbor, Kerala, India	*E. brucus*	1	Akhilesh et al., 2013 [17]
2009	Mediterranean Sea	Sea of Marmara, Türkiye	*E. brucus*	1	Kabasakal & Bilecenoglu, 2014 [23]
2009	Mediterranean Sea	Est Nile Delta, Egypt	*E. brucus*	48	Mancusi et al., 2020 [45]
2009	Mediterranean Sea	West Nile Delta, Egypt	*E. brucus*	27	Mancusi et al., 2020 [45]
2010	Indian Ocean	Karachi Fish Harbor, Pakistan	*E. brucus*	2	Moazzam & Osmany, 2021 [71]
2010	Indian Ocean	Kerala, India	*E. brucus*	1	Accession HM467790
2010	Indian Ocean	Kerala, India	*E. brucus*	1	Accession HM239653
2010	Indian Ocean	Oman	*E. brucus*	1	Straube et al., 2010 [36]; Naylor et al., 2012a [31]
2010	Mediterranean Sea	Aegean Sea, Türkiye	*E. brucus*	1	Mollen & Iglésias, 2023 [6]
2010	Mediterranean Sea	Sea of Marmara	*E. brucus*	1	Kabasakal & Bilecenoglu, 2014 [23]
2010	Mediterranean Sea	Türkiye	*E. brucus*	1	Mancusi et al., 2020 [45]
2008–2011	Indian Ocean	Kochi Fisheries Harbor, Kerala, India	*E. brucus*	431	Akhilesh et al., 2013 [17]
2009–2011	Indian Ocean	Kochi Fisheries Harbor, Kerala, India	*E. brucus*	3679	Akhilesh et al., 2020 [16]
2012	Atlantic Ocean	Louisiana, Gulf of Mexico, USA	*E. brucus*	1	Naylor et al., 2012b [30]
2012	Atlantic Ocean	D. F. Venezuela	*E. brucus*	1	Fariña et al., 2014 [29]
2012	Atlantic Ocean	La Tortuga, Venezuela	*E. brucus*	1	Mollen & Iglésias, 2023 [6]
2008–2013	Pacific Ocean	New Zealand	*E. brucus*	2	Francis, 2015 [61]
2013	Mediterranean Sea	Aegean Sea	*E. brucus*	1	Kabasakal & Bilecenoglu, 2014 [23]
2013	Mediterranean Sea	Izmir, Türkiye	*E. brucus*	1	Mancusi et al., 2020 [45]
2013	Mediterranean Sea	Türkiye	*E. brucus*	1	Mancusi et al., 2020 [45]
2000–2014	Mediterranean Sea	Türkiye	*E. brucus*	24	Kabasakal & Bilecenoglu, 2014 [23]
2014	Atlantic Ocean	Colombia, Atlantic	*E. brucus*	5	Anguila et al., 2016 [73]
2014	Indian Ocean	Kerala, India	*E. brucus*	1	Accession KJ749661
2014	Indian Ocean	Salalah, Oman	*E. brucus*	1	Al-Shajibi et al., 2014 [70]
2015	Indian Ocean	Kerala, India	*E. brucus*	1	Accession KR149154
2016	Indian Ocean	Kerala, India	*E. brucus*	1	Accession N. KU936195
2017	Mediterranean Sea	Off Şarköy, NW Sea of Marmara, Türkiye	*E. brucus*	1	Mancusi et al., 2020 [45]
2018	Atlantic Ocean	Namibia	*E. brucus*	4	Mollen & Iglésias, 2023 [6]
2018	Indian Ocean	Market in Sri Lanka	*E. brucus*	3	Fernando et al., 2019 [32]
2021	Indian Ocean	Bandar Al Khairan, Muscat, Oman	*E. brucus*	1	Morales-Avila et al., 2023 [69]
2021	Mediterranean Sea	Türkiye	*E. brucus*	17	Kabasakal et al., 2023 [24]
2022	Atlantic Ocean	Venezuela, D. F.	*E. brucus*	1	Ehemann & Zambrano-Vizquel, 2023 [51]
2022	Indian Ocean	Sibolga, Indonesia	*E. brucus*	1	Fahmi, 2022 [65]
2022	Mediterranean Sea	Türkiye	*E. brucus*	1	Mancusi et al., 2020 [45]
2023	Indian Ocean	Oman	*E. brucus*	1	Morales-Avila et al., 2023 [69]
NA	Atlantic Ocean	Rio Grande do Sul, Brazil	*E. brucus*	1	Mollen & Iglésias, 2023 [6]
NA	Indian Ocean	Oman	*E. brucus*	1	Naylor et al., 2012b [30]
NA	Mediterranean Sea	Albania	*E. brucus*	1	Soldo & Bakiu, 2021 [19]
NA	Mediterranean Sea	Nice, France	*E. brucus*	1	Mollen & Iglésias, 2023 [6]
1956	Pacific Ocean	Guadalupe Island	*E. cookei*	1	Long et al., 2011 [54]
1966	Pacific Ocean	Talara, Peru	*E. cookei*	2	Long et al., 2011 [54]
1972	Pacific Ocean	Puntarenas, Golfo de Nicoya, Costa Rica	*E. cookei*	1	Long et al., 2011 [54]
1972	Pacific Ocean	Northern California	*E. cookei*	19	Long et al., 2011 [54]
1972	Pacific Ocean	Southern California	*E. cookei*	7	Long et al., 2011 [54]
1973	Pacific Ocean	Cabo Blanco, Nicoya, Costa Rica	*E. cookei*	2	Long et al., 2011 [54]
1973	Pacific Ocean	Puntarenas, Golfo de Nicoya, Costa Rica	*E. cookei*	1	Long et al., 2011 [54]
1974	Pacific Ocean	Quepos, Costa Rica	*E. cookei*	1	Long et al., 2011 [54]
1974	Pacific Ocean	Gulf of California	*E. cookei*	2	Long et al., 2011 [54]
1975	Pacific Ocean	Golfo de California, Mexico	*E. cookei*	1	Ramos & Aguirre, 1975 [55]
1977	Pacific Ocean	Coast of Mazatlan, Sinaola, Mexico	*E. cookei*	1	Alvares-Leon & Castro-Aguirre, 1983 [56]
1959–1980	Pacific Ocean	Hawaii	*E. cookei*	13	Long et al., 2011 [54]
1980	Pacific Ocean	Cabo Blanco, Nicoya, Costa Rica	*E. cookei*	1	Long et al., 2011 [54]
1983	Pacific Ocean	Oregon	*E. cookei*	1	Long et al., 2011 [54]
1987	Pacific Ocean	SE Gulf of California, Mexico	*E. cookei*	1	Ruiz-Campos et al., 2010 [57]
1982–1992	Pacific Ocean	Hawaii	*E. cookei*	NA	Long et al., 2011 [54]
1992	Pacific Ocean	Monterey Bay, California	*E. cookei*	1	Bernardi & Powers, 1992 [58]
1992	Pacific Ocean	Northern California	*E. cookei*	96	Long et al., 2011 [54]
1994	Pacific Ocean	Guadalupe Island	*E. cookei*	2	Long et al., 2011 [54]
1994	Pacific Ocean	Southern California	*E. cookei*	7	Long et al., 2011 [54]
1995	Pacific Ocean	Galapagos Islands	*E. cookei*	3	Long et al., 2011 [54]
1996	Pacific Ocean	Baja California, Mexico	*E. cookei*	1	Galvàn-Magana et al., 1996 [59]
1997	Pacific Ocean	Chile	*E. cookei*	1	Long et al., 2011 [54]
1998	Pacific Ocean	Gulf of California	*E. cookei*	4	Long et al., 2011 [54]
1998	Pacific Ocean	Socorro Island	*E. cookei*	2	Long et al., 2011 [54]
1999	Pacific Ocean	Southwestern Mexico	*E. cookei*	1	Long et al., 2011 [54]
1999	Pacific Ocean	Michoacan Coast, Mexico	*E. cookei*	1	Aguirre et al., 2002 [60]
2000	Pacific Ocean	Southern California	*E. cookei*	1	Long et al., 2011 [54]
2000	Pacific Ocean	NW Tasmania, Australia	*E. cookei*	1	Bold System FMVIC514-08
2003	Pacific Ocean	W California, Mexico	*E. cookei*	1	Ruiz-Campos et al., 2010 [57]
2003	Pacific Ocean	Tasmania, Australia	*E. cookei*	1	Ward et al., 2008 [68]
2004	Pacific Ocean	Tasmania, Australia	*E. cookei*	1	Bold System FMVIC173-08
2005	Pacific Ocean	Cocos Island	*E. cookei*	46	Long et al., 2011 [54]
2005	Pacific Ocean	El Salvador	*E. cookei*	2	Long et al., 2011 [54]
2005–2006	Pacific Ocean	Northern California	*E. cookei*	25	Long et al., 2011 [54]
2006	Pacific Ocean	Hawaii, USA	*E. cookei*	1	A.N. DQ521002
2010	Pacific Ocean	Costa Rica	*E. cookei*	1	Accession N. OP035186
2010	Pacific Ocean	Gulf of California	*E. cookei*	1	Long et al., 2011 [54]
2010	Pacific Ocean	W. Baja California	*E. cookei*	1	Long et al., 2011 [54]
2012	Pacific Ocean	Queensland, Australia	*E. cookei*	1	Naylor et al., 2012b [30]
2008–2013	Pacific Ocean	New Zealand	*E. cookei*	26	Francis, 2015 [61]
2016	Pacific Ocean	Japan	*E. cookei*	1	Accession N. LC146082
2016	Pacific Ocean	South Australia	*E. cookei*	1	Bertozzi et al., 2016 [62]
2017	Pacific Ocean	Gulf of Tribugà, N Chocò, Colombia	*E. cookei*	1	Navia et al., 2023 [28]
2018	Pacific Ocean	Malpelo Island, Colombia	*E. cookei*	1	Bessudo et al., 2021 [63]
2019	Pacific Ocean	Chile	*E. cookei*	1	A.N. MK990519
2019	Pacific Ocean	Chile	*E. cookei*	1	A.N. MK990518
2021	Pacific Ocean	Nuqui, Choco, Colombia	*E. cookei*	1	Navia et al., 2023 [28]
2021	Pacific Ocean	Bahia de Los Angeles, Mexico	*E. cookei*	1	Rosales-Vasquez et al., 2023 [64]
2022	Indian Ocean	Muncar, Indonesia	*E. cookei*	1	Fahmi, 2022 [65]
2022	Pacific Ocean	San Juan, Charambirà, Colombia	*E. cookei*	1	Navia et al., 2023 [28]
NA	Pacific Ocean	Chile	*E. cookei*	1	Long et al., 2011 [54]
NA	Pacific Ocean	Chile	*E. cookei*	1	Long et al., 2011 [54]
NA	Pacific Ocean	Chile	*E. cookei*	2	Long et al., 2011 [54]
NA	Pacific Ocean	Chile	*E. cookei*	1	Long et al., 2011 [54]
NA	Pacific Ocean	Nazca Ridge, Peru–Chile Trench	*E. cookei*	2	Long et al., 2011 [54]
NA	Pacific Ocean	Nazca Ridge, Peru–Chile Trench	*E. cookei*	NA	Long et al., 2011 [54]
NA	Pacific Ocean	NE Sidney	*E. cookei*	1	A.N. HQ956324
NA	Pacific Ocean	Mexico	*E. cookei*	1	Del Moral-Flores et al., 2015 [66]
NA	Pacific Ocean	NW Mexico	*E. cookei*	1	Melendez & Garayzar, 1998 [67]

## Data Availability

Data are contained within the article.

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
