# Peer review of "Genetic and Distribution Data of the Bramble Shark Echinorhinus brucus (Bonnaterre, 1788) and the Prickly Shark Echinorhinus cookei Pietschmann, 1928 to Better Reconstruct Their Conservation Status"

_animals, 2024, doi:10.3390/ani14070993_

Round 1

Reviewer 1 Report

Comments and Suggestions for Authors

RE:

 Genetic and distribution data of the bramble shark Echinorhinus brucus (Bonnaterre, 1788) to better reconstruct conservation 3 status

By: Battiata et al.

This manuscript discusses the vulnerability of elasmobranch species, such as sharks and rays, to various anthropogenic stressors like fishing, habitat loss, and climate change, which puts them at risk of extinction. Specifically, it focuses on Echinorhinus brucus, a deep-sea shark classified as "Endangered" with limited available information about its distribution and genetic structure. The study examined mitochondrial DNA sequences from both E. brucus and its closely related species, Echinorhinus cookei, and conducted phylogenetic analyses. Interestingly, E. brucus sequences from the Indian Ocean clustered with E. cookei rather than with Atlantic E. brucus, suggesting possible misidentification or genetic connectivity between the two species. The study also reviewed historical and recent captures, revealing current local extinctions in certain areas like the North Sea and the western Mediterranean Sea. It identified specific areas of higher E. brucus abundance in the Atlantic Ocean (Brazil) and the Indian Ocean (India), which could be focal points for conservation efforts, especially considering the genetic differences highlighted by molecular analysis.

The manuscript is in my opinion interesting, and the phylogenetic analyses have been done in a competent way, therefore I do not have any concern about the Bayesian approaches utilised.

A minor comment, figure 1 does not need to be filled with irrelevant details as the altitude (green, brown and blue colours).

I would also like if the authors could give more details when…writing…Despite the results could be affected by the small number of data, our analyses support the taxonomic criticisms within the genus already been noticed by other authors,  which, in some cases, applied the Neighbor-Joining of p-distances [27–30,47]. In particular, Fernando et al. [30] exhibited the same topology of the present trees……….if you could give some more details about the topology and the support values of the Neighbor-Joining of p-distances that would be great.

I would also like to speculate a little bit more about the sentence….. Molecular results for E. cookei (Figure 3) show that haplotypes from Australia do not  cluster together. One of the two specimens from Tasmania is strictly grouped with the  ones from Hawaii and from Queensland. The other two from New South Wales and Tasmania cluster with individuals from Chile and Costa Rica. The actual knowledge of the reproductive strategies E. cookie is insufficient to assert some considerations about dispersal. ……it is okay to write that…. The actual knowledge of thereproductive strategies E. cookie is insufficient…….but could you speculate about the reason for this result?

Author Response

Dear Editor

please, find attached the revised version of the ms.

We thank the reviewers for their help in improving the manuscript; we found their recommendations appropriate and helpful.

Below are our answers (in bold) point by point.

Best regards

Sabrina Lo Brutto

Answers to Rev.1

This manuscript discusses the vulnerability of elasmobranch species, such as sharks and rays, to various anthropogenic stressors like fishing, habitat loss, and climate change, which puts them at risk of extinction. Specifically, it focuses on Echinorhinus brucus, a deep-sea shark classified as "Endangered" with limited available information about its distribution and genetic structure. The study examined mitochondrial DNA sequences from both E. brucus and its closely related species, Echinorhinus cookei, and conducted phylogenetic analyses. Interestingly, E. brucus sequences from the Indian Ocean clustered with E. cookei rather than with Atlantic E. brucus, suggesting possible misidentification or genetic connectivity between the two species. The study also reviewed historical and recent captures, revealing current local extinctions in certain areas like the North Sea and the western Mediterranean Sea. It identified specific areas of higher E. brucus abundance in the Atlantic Ocean (Brazil) and the Indian Ocean (India), which could be focal points for conservation efforts, especially considering the genetic differences highlighted by molecular analysis.

The manuscript is in my opinion interesting, and the phylogenetic analyses have been done in a competent way, therefore I do not have any concern about the Bayesian approaches utilised.

A minor comment, figure 1 does not need to be filled with irrelevant details as the altitude (green, brown and blue colours).

  • The map of Figure 1 has been changed and revised in a simpler graphical layout, without altitude.

I would also like if the authors could give more details when…writing…Despite the results could be affected by the small number of data, our analyses support the taxonomic criticisms within the genus already been noticed by other authors,  which, in some cases, applied the Neighbor-Joining of p-distances [27–30,47]. In particular, Fernando et al. [30] exhibited the same topology of the present trees……….if you could give some more details about the topology and the support values of the Neighbor-Joining of p-distances that would be great.

  • We changed the text and gave more details about the statistics and topology of the cited analyses:

In particular, Fernando et al. [30] and Naylor et al. [44] exhibited the same topology of the present trees using different methodologies. In particular, the first author used a Neighbor-Joining with the uncorrected p-distance model and the second one a Bayesian approach using GTR+I+Γ model, both using NADH2 as marker.

 I would also like to speculate a little bit more about the sentence….. Molecular results for E. cookei (Figure 3) show that haplotypes from Australia do not  cluster together. One of the two specimens from Tasmania is strictly grouped with the  ones from Hawaii and from Queensland. The other two from New South Wales and Tasmania cluster with individuals from Chile and Costa Rica. The actual knowledge of the reproductive strategies E. cookie is insufficient to assert some considerations about dispersal. ……it is okay to write that…. The actual knowledge of the reproductive strategies E. cookie is insufficient…….but could you speculate about the reason for this result?

  • We changed the sentence. Regarding the pattern scored in E. cookei, we really do not have any information that can explain such a dichotomy. The actual knowledge of the reproductive strategies of E. cookie is insufficient to assert any consideration but our dataset could serve as a stimulus for new studies on the possible dispersal behaviour of this species.

Reviewer 2 Report

Comments and Suggestions for Authors

The idea is good, however the study needs to be restructured, rethinking its objectives and scope.

The title should be changed to include E. cookei because it is frequently mentioned throughout the manuscript. 

I noticed confusion in several concepts and it was not clear to me the true purpose of the study, see the comments throughout the manuscript

Comments on the Quality of English Language

Language needs to be revised

Author Response

Dear Editor

please, find attached the revised version of the ms.

We thank the reviewers for their help in improving the manuscript; we found their recommendations appropriate and helpful.

Below are our answers (in bold) point by point.

Best regards

Sabrina Lo Brutto

Answer to Rev.2

The idea is good, however the study needs to be restructured, rethinking its objectives and scope.

  • We planned the present study to assess the genetic diversity of the two species and contextually their distribution and to synthesize the huge amount of information from a high number of sources. The paper describes the existence of reproductively isolated entities that need a taxonomic revision. The genetic pattern is also congruent with the distribution of the two species and the diverse entities within E.brucus as their ranges do not overlap.

On the whole, the paper is a baseline for future investigations.

The title should be changed to include E. cookei because it is frequently mentioned throughout the manuscript.

  • We agree with the Reviewer, thus we changed the title in:

Genetic and distribution data of two sharks, the bramble shark Echinorhinus brucus (Bonnaterre, 1788) and the prickly shark Echinorhinus cookei Pietschmann 1928, to better reconstruct their conservation status

I noticed confusion in several concepts and it was not clear to me the true purpose of the study, see the comments throughout the manuscript

We followed some of the Rev.’s suggestions indicated in the file.

We did not accept the following comments:

Line12 “Sharks and rays are threatened by human activities like fishing and climate change. Many of these species are at risk of extinction.” Rev’s comment: Which species?? shark and rays are groups of chondrichthyasn that includes several species. But shark and rays are not species. Please modify

  • The word species is here used in a general sense, not regarding the taxonomic rank.

We know that sharks and rays are not species, however we can use the word “species” to indicate an animal group.

Line 23 “The species could include two different entities not fully detected by scientists yet.” Rev’s comment: ??= diferent morphs, taxa, species??

  • We indicate entities in a general phylogenetic or phylogeographic sense, thus without indicating species or morphs..etc.

Line 26 “In this sense, the elasmobranch populations appear to be at risk of extinction in many cases.” Rev’s comment: Populations and species are different, a population

  • The comment is not clear.

Line 39 “The worldwide distribution of E. brucus excludes the Pacific Ocean area,” Rev’s comment: Apparently only from the eastern Pacific, but exist a record from NW Pacific sensu GBIF

  • See our answer below.

Line 92 “In 2014, a 92 female was captured in the Caribbean waters of Venezuela and the particular pattern of 93 dermal denticles did not allow a confident identification due to the shape intermediate” Rev’s comment: So, this is not a diagnostic character?

  • We wrote that this peculiar case was a questionable identification of a single specimen, and it was a record extrapolated by literature to be reported in our paper.

Line 211 “our analyses support the taxonomic criticisms within the genus already been noticed by other authors” Rev’s comment: what is it? What can you comment on this?

  • The existence of two lineages within E.brucus and its paraphyly.

Line 227 “There is no evidence of the presence of E. brucus in the Pacific Ocean, corroborated” Rev’s comment: https://www.gbif.org/species/2421209

  • We checked the GBIF dataset

https://www.gbif.org/occurrence/search?has_coordinate=true&has_geospatial_issue=false&occurrence_status=present&taxon_key=2421209

The record is a specimen identified as E.spinosus collected in 1907 that should be stored at the Museum of Comparative Zoology, Harvard University. The collection preserved in such a museum has been included in Mollen & Iglésias 2023, whose records we included in our dataset.

According to Mollen & Iglésias 2023, the record has not been validated and we could not add it.

Reviewer 3 Report

Comments and Suggestions for Authors The authors performed molecular phylogenetic analysis of echinorhiniform sharks with mitochondrial genes and analyzed its results with the geographical distribution. Although the size of the dataset is very modest (without any sequences obtained by the authors), I found the large value of publishing this study that also includes the record of the specimen and important literature. I support the publication of this manuscript, but only after the points listed below are taken care of in revising the manuscript.   Figure 3 - This is the most crucial point. Why doesn't the tree in Figure 3 have variable branch lengths?   Figures 1 and 3 - The use of a single genus or a single species as outgroup cannot be justified in rooting the phylogeny of such an isolated lineage of species. Consider using species from multiple orders of Squalimorphii.   The title should also include the species name E. cookei. I do not know why it is left out.   Line 73 - The phrase 'bycatches fisheries' in Introduction looks wrong. Maybe 'in' should be necessary between the words?   Line 170 - Skip the word 'very'.   Figure 1 - The letters in the legend of the figure are too small to read. This also applies to Figure 3.   Line 211 - The grammatical usage of the word 'despite' should be reconsidered.

Comments on the Quality of English Language

Please refer to the comments.

Author Response

Dear Editor

please, find attached the revised version of the ms.

We thank the reviewers for their help in improving the manuscript; we found their recommendations appropriate and helpful.

Below are our answers (in bold) point by point.

Best regards

Sabrina Lo Brutto

Answer to Rev.3

The authors performed molecular phylogenetic analysis of echinorhiniform sharks with mitochondrial genes and analyzed its results with the geographical distribution. Although the size of the dataset is very modest (without any sequences obtained by the authors), I found the large value of publishing this study that also includes the record of the specimen and important literature. I support the publication of this manuscript, but only after the points listed below are taken care of in revising the manuscript.  

Figure 3 - This is the most crucial point. Why doesn't the tree in Figure 3 have variable branch lengths?

  • Thanks for the question, the figure 3 shows a Bayesian molecular clock analysis in fact, as reported in the materials and methods, we use a mutational rate of 0.07*10^-8 mutations per year. The result of this kind of analysis is a time-tree on where all the tips represent the present and nodes show the age of the splits; that’s why all the tips are aligned. Unfortunately, the small number of individuals and a very poor and fragmented information about the evolutionary story of this genus (unclear fossils, lack of slow mutational rate sequences, lack of any evaluation of saturated sites) have as a result a tree with too stretched branches and nodes along the past but the correct handling of Bayesian programs gives a very strong support about relations between the different lineages even if the time of the various splits is not coherent.
  •  

Figures 1 and 3 - The use of a single genus or a single species as outgroup cannot be justified in rooting the phylogeny of such an isolated lineage of species. Consider using species from multiple orders of Squalimorphii. 

  • We used 3 individuals of P. cirratus for COI analyses to have a proportionated number of sequences if compared with the target ones and one of S. crassispinus for NADH2. We evaluated unnecessary the use of more individuals (also from other families) because the deep relationship between Echinorhiniformes, Squaliphormes, Pristiophoriformes and Hexanchiformes is still debated but anyway Echinorhinidae species are considered a monophyletic group and recently uploaded at order level (i.e. Echinorhiniformes). As found in all our analysis, roots have always been found with 100 bootstrap values.

Naylor, G. J., Ryburn, J. A., Fedrigo, O., & Lopez, J. A. (2005). Phylogenetic relationships among the major lineages of modern elasmobranchs. Reproductive biology and phylogeny, 3(1), 25.

Adnet, S., & Cappetta, H. (2001). A palaeontological and phylogenetical analysis of squaliform sharks (Chondrichthyes: Squaliformes) based on dental characters. Lethaia, 34(3), 234-248.

Flammensbeck, C. K., Pollerspöck, J., Schedel, F. D., Matzke, N. J., & Straube, N. (2018). Of teeth and trees: A fossil tip‐dating approach to infer divergence times of extinct and extant squaliform sharks. Zoologica Scripta, 47(5), 539-557.

The title should also include the species name E. cookei. I do not know why it is left out.  

  • The tile has been changed

Line 73 - The phrase 'bycatches fisheries' in Introduction looks wrong. Maybe 'in' should be necessary between the words?   Line 170 - Skip the word 'very'.   Figure 1 - The letters in the legend of the figure are too small to read. This also applies to Figure 3.   Line 211 - The grammatical usage of the word 'despite' should be reconsidered.

  • Done

Round 2

Reviewer 2 Report

Comments and Suggestions for Authors

The manuscript was improved in several aspects. However some inconsistencies still persist in this wording.

The authors must be clear about the difference between a taxnomic review and one like the study presented here.

The main weakness of he study is the analysis of the results and its discussion . Particularly, the discussion lacks background to supportwhats is writen by the authors.

For example, it is not clear to mehoe the authors confirm the identity of E. brucus using genetic sequences and Blast, but argue that could be a subspecies or a new "entity" as they mention.

See comments in the attached documment

Comments on the Quality of English Language

The manuscript needs to be reviewed

Author Response

Dear Editor

please, find attached the second revised version of the ms.

We thank the reviewer for her/his help in improving the manuscript; we found her/his recommendations appropriate and helpful; thus we followed them.

We also revised the manuscript strictly following the Instructions of the journal and added the Conclusion section which is mandatory.

Below are our answers (in bold) point by point.

We attached the revised version.

Best regards

Sabrina Lo Brutto

Answers to Rev.2

Line 105 (of the previous version) to assess the systematic of the genus you indeed include more representative species of the other related genera of the family and or the order

  • Changed accordingly

Line 211 (of the previous version) The use of et al., implies a plural condition

  • Changed accordingly

Line 117(of the previous version)   It is interesting to know, How from genetic sequences, you confirmed the morphological similarity with the species?

  • We verified through pictures of individuals

Line 218 (of the previous version) Where in the text is such a distance mentioned?

  • We added the data regarding the p-distance of COI using the Gamma Distributed Rates among Site and Gamma parameter 4.00. The results are shown in Table 1.

Line 220 (of the previous version) What type of barriers may be present in the marine environment? Mention possible examples for mesopelagic and deep waters

  • We specified: barriers to gene flow

Line 230 (of the previous version) This paragraph is a very poor discussion of the zoogeography of the species.

  • The paragraph is related with the next; we have brought them together.

Line 232 (of the previous version) northern California is not Central America

  • changed accordingly

Line 233 (of the previous version) So, how were the genetic samples taken?

  • Individuals genetically sampled are different in individuals from the ones observed in submergible missions. We rewrited the sentence to make it clearer.

Line 238 (of the previous version)  Okay, but what can be discussed in this regard? This statement is a paraphrase of the results but no possible explanation is given as to why we have this topology.

  • Actually, the data set is very small and the presence of independent mutation in Australian sequences can have affected the final tree resulting in a wrong topology. Another possible explanation might be due to the dispersal of individuals which were born in different areas or the old biogeographical relations. So as we will discuss in the comment for line 241, each explanation should be considered speculation.

Line 241 (of the previous version) The explanation must be given based on the biogeography of the species. I do not believe that the dispersalist argument is sufficient to explain the current distribution, but the old relationship between biogeographical areas or regions does

  • The actual knowledge of the reproductive strategies of E. cookei is insufficient to assert any consideration and all explanations should be considered speculation. The dataset is too small and a group of individuals from the same areas cannot be considered a real genetic population so we just described what we found: nearest individuals do not cluster together and we highlight how this could be interesting for future studies.

Line 246 According to your distribution and abundance map, I am not sure that there are correspondences between the georeferencing of the records and the upwelling zones around the world

  • These conditions are reported by different authors (Eheman and Zambrano-Vizquel, 2023 and ref. therein)

Attached is the revised manuscript.
